# Monocytes in the Characterization of Pain in Palliative Patients with Severe Dementia—A Pilot Study

**DOI:** 10.3390/ijms241310723

**Published:** 2023-06-27

**Authors:** Hugo Ribeiro, Raquel Alves, Joana Jorge, Ana Cristina Gonçalves, Ana Bela Sarmento-Ribeiro, Manuel Teixeira-Veríssimo, José Paulo Andrade, Marília Dourado

**Affiliations:** 1Community Support Team in Palliative Care—Group of Health Centers Gaia, 4400-043 Vila Nova de Gaia, Portugal; 2Faculty of Medicine, University do Porto, 4200-219 Porto, Portugal; 3Faculty of Medicine, University of Coimbra, 3004-304 Coimbra, Portugalmdourado@fmed.uc.pt (M.D.); 4Coimbra Institute for Clinical and Biomedical Research (iCBR)—Group of Environment Genetics and Oncobiology (CIMAGO), FMUC, 3004-304 Coimbra, Portugal; 5Center for Innovative Biomedicine and Biotechnology (CIBB), 3004-304 Coimbra, Portugal; 6Laboratory of Oncobiology and Hematology (LOH), Faculty of Medicine, University of Coimbra, University Clinics of Hematology and Oncology, 3004-304 Coimbra, Portugal; 7Hematology Service, Centro Hospitalar e Universitário de Coimbra (CHUC), 3000-075 Coimbra, Portugal; 8CINTESIS@RISE, Faculty of Medicine, University of Porto, 4200-219 Porto, Portugal; 9Unit of Anatomy, Department of Biomedicine, Faculty of Medicine, University of Porto, 4200-219 Porto, Portugal

**Keywords:** monocytes, biomarkers, chronic pain, dementia, pain characterization

## Abstract

In assessing and managing pain, when obtaining a self-report is impossible, therapeutic decision-making becomes more challenging. This study aimed to investigate whether monocytes and some membrane monocyte proteins, identified as a cluster of differentiation (CD), could be potential non-invasive peripheral biomarkers in identifying and characterizing pain in patients with severe dementia. We used 53 blood samples from non-oncological palliative patients, 44 patients with pain (38 of whom had dementia) and 0 without pain or dementia (controls). We evaluated the levels of monocytes and their subtypes, including classic, intermediate, and non-classic, and characterized the levels of specific phenotypic markers, namely CD11c, CD86, CD163, and CD206. We found that the relative concentrations of monocytes, particularly the percentage of classic monocytes, may be a helpful pain biomarker. Furthermore, the CD11c expression levels were significantly higher in patients with mixed pain, while CD163 and CD206 expression levels were significantly higher in patients with nociceptive pain. These findings suggest that the levels of monocytes, particularly the classic subtype, and their phenotype markers CD11c, CD163, and CD206 could serve as pain biomarkers in patients with severe dementia.

## 1. Introduction and Objectives

Pain is a multidimensional experience that can significantly impair a patient’s quality of life. It can be classified according to the underlying cause, including nociceptive pain resulting from tissue damage, neuropathic pain caused by nerve injury, and nociplastic pain resulting from altered pain modulation [1]. As multiple factors influence pain, it is essential to characterize it accurately to treat chronic pain patients with success [2].

Patient self-reporting is regularly used to characterize pain. However, it is subject to individual interpretation, which can be a problem for proper pain characterization. To overcome this subjectivity, multidimensional hetero-assessment scales have been developed, particularly for patients with dementia or others unable to characterize their pain [3,4,5]. However, this approach is insufficient for this type of patient, and there is an urgent need for a more objective characterization using non-invasive biomarkers.

Routine peripheral blood parameters, including white blood cell counts and mean platelet volume (MPV), have been identified as potential diagnostic, prognostic, and predictive response markers in inflammatory and central nervous system diseases, such as cerebral hemorrhage and dementia in elderly patients [6]. Further, blood plasma, blood cells, skin fibroblasts, and peripheral blood vessels were considered diagnostic biomarkers for Alzheimer’s disease [7]. Our previous studies show that some platelet cluster of differentiation (CD) levels could be valuable as pain biomarkers, particularly for pain subtype classification and pain intensity characterization [8].

Monocytes are a type of white blood cell crucial in promoting and resolving inflammation and are implicated in several inflammatory diseases, such as cardiovascular diseases, diabetes, obesity, and metabolic syndrome [9,10]. There are three main subsets of monocytes, classical (~85%), non-classical (~10%), and intermediate (~5%), which are characterized by their expression levels of CD14 and CD16 [10,11]. These subsets play distinct roles in homeostasis, inflammation, and various diseases such as rheumatoid arthritis, complex regional pain syndrome, cancer, tuberculosis, and HIV [11,12,13,14]. Studies have reported an increased proportion of intermediate monocytes in various inflammatory states, including cardiovascular disease [9,10]. Circulating monocytes correlate with several diseases that cause pain [15,16,17]. However, statistical robustness for this correlation with pain and pain characterization is lacking.

This study aims to assess the levels of monocyte subsets (classical, non-classical, and intermediate) [10,16] and the transmembrane proteins CD11c, CD86, CD163, and CD206, which are related to various monocyte functions, including immune response during infection or inflammation [18,19,20,21] in non-oncological palliative patients. Furthermore, this study aims to assess the potential of these membrane proteins in characterizing pain and serving as non-invasive biomarkers for patients who cannot self-report their pain, especially those with advanced dementia.

## 2. Results

We initially selected 95 patients. However, the legal representatives of five patients refused to participate in this study. Seventeen patients were excluded from blood sample collection due to their fragile condition, hypovolemia, and difficult venous access. Also, 20 patients were in their last days of life, and we decided to avoid blood collection in this clinical condition.

We obtained blood samples from 53 patients with an average age of 74.8 years old, a minimum of 29, and a maximum of 98 years old. Most of the patients were female [n = 39 (73.6%)]. Nine patients had no pain, and forty-four suffered from chronic pain. We had no patients diagnosed with autoimmune diseases such as inflammatory bowel disease, multiple sclerosis, psoriasis, or other conditions.

Of the 44 patients with chronic pain, 38 also had severe dementia, making assessing their pain intensity and type difficult. As a result, we relied on the PAINAD scale to identify patients who likely had uncontrolled pain. Patients with PAINAD ≥ 5 (present in 32 patients) are those who probably have uncontrolled pain [22]. Among the 44 patients with pain, 19 were receiving opioid treatment (15 with dementia and 4 without dementia). We saw no significant differences between the prevalence of pain in males and/or females. When we compared the monocyte phenotype between different clinical features such as renal function, hepatic function, nutritional status, and biotype (weight, height), we did not observe any statistically significant difference.

We had six patients with pain without dementia and nine patients without pain and without dementia that were considered controls, as shown in Table 1.

### 2.1. Monocyte Characterization in Patients with and without Chronic Pain

In patients with pain, we observed a significant increase in the percentage of total monocytes (10.27% vs. 4.19%, *p* = 0.025), in particular in the classic monocytes (91.1% vs. 85.3%, *p* = 0.003), and we saw a decrease in the intermediate (5.7% vs. 3.1%, *p* = 0.008) and non-classic monocytes (5.53% vs. 2.79%, *p* = 0.011) compared with patients without pain (Figure 1 and Table 2).

When we evaluated, in each monocyte subset, the expression of CD11c, CD163, and CD206, we did not find any statistical differences in these markers in intermediate and non-classic monocytes between patients with and without pain. However, in total and classic monocytes, we found statistically significant differences in CD206 and CD163, when comparing these biomarkers with the presence of pain, as shown in Table 2. We discovered a significant rise in the proportion of monocytes expressing CD206 in patients experiencing pain, as opposed to those without pain (59.36% vs. 14.08%, *p* = 0.047). On the other hand, we detected, in these patients, a decrease in the expression levels of CD206 (469.01 vs. 534.29 MFI, *p* = 0.019), CD163 (2378.85 vs. 3061.99 MFI, *p* = 0.05), and in the ratio of CD163/CD206 (4.81 vs. 9.5, *p* = 0.004) in monocytes (Table 2). In classical monocytes, all of the same protein receptors are decreased in patients with pain compared to those without pain (percentage of cells expressing CD163—95.6 vs. 88.0 MFI, *p* = 0.01, CD206—11.2% vs. 6.7%, *p* = 0.039, and ratio CD163/CD206—11.1% vs. 6.5%, *p* = 0.038 and expression levels of CD206 were 503.4 vs. 472.9 MFI, *p* = 0.021).

No differences between controlled and uncontrolled pain patients were observed in the studied biomarkers.

### 2.2. Monocytes and Gender

When comparing the biomarkers and the gender of patients, we found significant differences between the percentage of cells expressing CD86 and the ratio CD11c/CD86, which have higher values in men (30.0 vs. 20.5 MFI, *p* = 0.026 and 28.37 vs. 20.1 MFI, *p* = 0.047).

### 2.3. Type of Pain and Monocyte Biomarkers

As presented in Table 3, we only detected statistically significant differences in the percentage of non-classic monocytes. The percentage was higher in patients with nociceptive pain (3.72% vs. 1.82%, *p* = 0.037) than in patients with mixed pain.

However, we found statistically significant differences when comparing types of pain and CD11c, CD163, and CD206.

The percentage of monocytes expressing CD11c is significantly higher in patients with mixed pain, while in patients with nociceptive pain, we observed an increase in the expression levels of CD163 and CD206 and the ratio CD163/CD206.

Concerning the monocyte subpopulations, there is an increase in the expression levels of CD11c in all subpopulations of monocytes in patients with mixed pain compared with those with nociceptive pain, in which the percentage of monocyte expression of this receptor is only changed in classic and intermediate monocytes (Table 3). Moreover, in non-classic monocytes, only CD11c is altered.

In patients with nociceptive pain, the expression levels of CD206 are significantly higher in classical and intermediate monocytes compared with patients with mixed pain (classic—478.16 vs. 469.28 MFI, *p* = 0.016; intermediate—575.94 vs. 517.74 MFI, *p* = 0.041), while CD163 is only changed in classic monocytes.

Therefore, while the levels of CD163 and CD206 are significantly higher in nociceptive pain, the levels of CD11c are significantly higher in mixed pain.

With multilogistic regression, we did not find any statistically relevant differences, so we did not perform multinominal regression. The specificity of the model was 88.2% and the sensibility was 77.8%. The model’s discriminative capacity was exceptional (0.905), with the area under the curve being statistically significant, as is shown in Figure 2.

### 2.4. Analgesic Drugs and Monocyte Biomarkers

We also analyzed if the treatment with opioids and paracetamol interfered with the monocyte biomarkers. Consequently, we conducted a comparison of monocyte biomarker levels between patients who were receiving treatment with opioids and paracetamol and those who were not. Our findings indicated that CD163 was the sole biomarker with statistical significance. In patients receiving opioid therapy, the percentage of monocytes expressing CD163 was lower than those not receiving opioids (values of total monocytes and the three subtypes), as shown in Table 4. These levels are higher in patients receiving paracetamol treatment, particularly in non-classical monocytes. No significant differences were found between the studied monocyte biomarkers and the equivalent dose of morphine.

### 2.5. Monocyte Characterization and Dementia

In order to correlate the phenotypic markers in monocytes with dementia, we analyzed the levels of different monocyte subsets and CD in patients with and without dementia of different subtypes.

As shown in Table 5, we found some significant differences between the biomarkers and dementia. One finding is that patients with dementia have a higher relative concentration of monocytes than those without dementia (11.08% vs. 4.73%, *p* = 0.037). Notably, patients with vascular dementia exhibited the lowest levels of relative concentrations of monocytes.

Moreover, in the monocytes of dementia patients, the expression levels of CD163, CD206, and the ratio of CD163/CD206 are lower than those detected in patients without dementia (CD163—3420.37 vs. 2108.14 MFI, *p* = 0.032; CD206—562.85 vs. 445.58 MFI, *p* = 0.001; CD163/CD206—9.2 vs. 4.1, *p* = 0.01). Moreover, the classic and intermediate monocytes of patients with dementia show a significant decrease in the percentage of monocytes expressing CD163, while the decrease in the expression levels of CD206 is mainly observed in classic and non-classic monocytes, when compared with patients without dementia (classic—528.2 vs. 456.2 MFI, *p* = 0.002; non-classic—752.9 vs. 447.3 MFI, *p* = 0.008) (Table 5).

When comparing biomarkers with types of dementia, there are statistically significant differences, particularly in the relative concentration of monocytes, as previously mentioned, and in the expression levels of CD206, which are highest in patients with vascular dementia (17.06% vs. 4.73%, *p* = 0.028 and 465.70 vs. MFI 419.85, *p* = 0.007, respectively).

When we analyzed the different subsets of monocytes in patients with different types of dementia, in intermediate monocytes, we did not observe significant differences among Alzheimer’s disease, vascular, and other types of dementia. Conversely, we observed a significant increase (*p* = 0.045) in the expression of CD206 in non-classical monocytes among patients with vascular dementia (MFI 561.1), whereas the lowest expression was observed in patients with Alzheimer’s disease (MFI 482.0). Our findings showed a significant decrease (*p* = 0.049) in the expression levels of CD86 (MFI 579.3) in classical monocytes of Alzheimer’s disease patients compared to those with other dementias (MFI 597.1). Moreover, the relative percentage of CD163 was significantly higher in Alzheimer’s patients than in those patients with vascular dementia (MFI 93.6 vs. MFI 84.9, *p* = 0.036).

### 2.6. Monocyte Characterization and Opioids and Other Analgesics Used to Control Pain in These Patients

When we compared the percentages of monocytes, and of the monocyte subsets characterized by the different CD, referred to before in patients receiving opioids and in patients that did not receive opioids, we did not find statistically significant differences between the two groups of patients. We had identical findings when comparing patients receiving other analgesics, such as paracetamol and/or NSAIDs, with patients without any pain treatment.

## 3. Material and Methods

For this study, we obtained clinical and individual data and blood samples from 53 palliative patients with non-oncological diseases, followed by a palliative care team between 1 September and 31 December 2021.

This study is an observational, analytical, cross-sectional, non-interventional investigation that utilizes the medical and nursing records of patients with chronic pain.

The North Regional Health Administration of Portugal (ARS Norte) and the Faculty of Medicine (FMUP) Ethics Committee accepted the research procedures. The research was conducted following the principles of the Declaration of Helsinki. In compliance with ethical guidelines on confidentiality, data anonymity, and voluntary withdrawal, the participants or their representatives provided informed consent before study enrollment.

We collected individual and clinical data by reviewing the records in the patient’s clinical files, which were then saved in a secure Excel sheet [23]. Each patient was assigned an alphanumeric code to ensure patient confidentiality, known only to the researcher. Following the European General Data Protection Regulation (GDPR), the Excel files will be eliminated upon the completion of the study and publication of the results. The data included age, gender, pain type and intensity, opioid and other analgesic drugs, such as nonsteroidal anti-inflammatory drugs (NSAIDs) and acetaminophen, and their corresponding doses. Additionally, we recorded whether the patient’s pain was controlled during blood sample collection.

We also obtained information regarding the diagnosis and type of dementia. In cases where patients had severe dementia, we utilized the Pain Assessment in Advanced Dementia Scale (PAINAD) [22] to differentiate those who may present uncontrolled pain. Based on the PAINAD scores, we categorized patients into three groups, scores below 5, scores between 5 and 7, and scores between 8 and 10, to investigate the potential correlation between these scores and mild, moderate, or severe pain, respectively. We administered the numeric pain scale for patients who could self-report their pain (Rodriguez, 2001).

To analyze the expression of monocyte biomarkers, we selected CD11c [24], CD86 [25], CD163 [26] and CD206 [27] both absolute and relative concentration. We conducted a receiver–operator characteristic (ROC) curve analysis to determine the area’s significance under the curve and establish a cut-off point for the relevant markers.

### 3.1. Evaluation of Monocytes Subsets by Flow Cytometry

We collected peripheral blood samples in EDTA tubes to analyze the monocytes. Monocytes were classified into three subsets based on the expression levels of CD14 and CD16. These subsets are the classical monocytes (CD14+CD16−), non-classical monocytes (CD14−CD16+), and intermediate monocytes (CD14+CD16+) [11]. Following a 15 min incubation in the dark at room temperature, erythrocytes were lysed using a BD. Pharm Lyse™ reagent according to the manufacturer’s protocol (BD Pharmingen, BD Biosystems, San Diego, CA, USA). Cells were run through a FACS Canto II flow cytometer (BD Biosystems), and at least 100,000 events were collected using FACS DIVA software (BD Biosystems). Data were studied with the Kaluza Software (Beckman Coulter, Jersey City, New Jersey), and the results are expressed using a total of monocytes (CD14+) and the percentage of each monocyte subset based on the cells’ positivity for CD14 and CD16 expression [monocytes classic (CD14+/CD16−), non-classic (CD14+/CD16++) and intermediates (CD14++/CD16+)].

### 3.2. Evaluation of Monocyte Subsets and Membrane Proteins Using Flow Cytometry

The monocyte analysis was performed in peripheral blood samples collected in EDTA tubes to identify the monocyte subsets and characterize the transmembrane protein receptors related to its recognized functions.

We stained 200 µL of whole blood with the following monoclonal antibodies (mAbs): anti-CD14 BD Pharmingen™ (APC), anti-CD16 BD Horizon™ (BV500) mAbs; anti-CD11c BD Pharmingen™ (FITC), anti-CD163 BD Pharmingen™ (PE), anti-CD86 BD Pharmingen™ (PerCP-Cy5.5), and anti-CD206 BD Horizon™ (BV421). Following a 15 min incubation in the dark at room temperature, erythrocytes were lysed using a BD. Pharm Lyse™ reagent according to the manufacturer’s protocol. Cells were run through a FACS Canto II flow cytometer (BD Biosystems), and at least 100,000 events were collected using FACS DIVA software (BD Biosystems). Data were studied with the Kaluza Software (Beckman Coulter), and the results were expressed in a percentage of positive cells for each marker and respective mean fluorescence intensity (MFI). Each marker was analyzed in the total monocyte population (CD14+). Additionally, a sub-analysis was performed based on monocyte classification, using CD14 and CD16 expression, into classical (CD14^+^CD16^−^), non-classical (CD14^−^CD16^+^), and intermediate (CD14^+^CD16^+^) monocytes. In each monocyte subset, the expressions of CD11c, CD163, and anti-CD206 were determined.

### 3.3. Statistical Analysis

We used SPSS (version 28.0 for Windows, IBM Corp., Armonk, NY, USA) for statistical analysis. Descriptive statistics measures such as absolute and relative frequencies, means, and standard deviations were employed for data analysis. Additionally, inferential statistics were also utilized. We set the level of significance (α) at ≤0.05 to reject the null hypothesis. The Student’s *t*-test for independent samples, one-way ANOVA, Mann–Whitney U test, Kruskal–Wallis test, and logistic, binary, and multinomial regressions were used. The normality of distribution was analyzed with the Kolmogorov–Smirnov test. The homogeneity of variances was evaluated with Levene’s test. Multicollinearity was analyzed with VIF and Tolerance. ROC analysis and the calculation of the cut-off value for the biomarkers, as well as the specificity, sensitivity, positive predictive, and negative predictive values, were performed.

We incorporated several variables into the analysis, including sex, age, type and intensity of pain, opioid type and dosage, and other analgesics like nonsteroidal anti-inflammatory drugs (NSAIDs) and paracetamol, as well as absolute (MFI) and relative concentration levels of CD11c, CD86, CD163, and CD206. In our analysis, we factored in the type of dementia, since it was prevalent among most patients, and controlled pain time for patients presenting with chronic pain but whose pain was not present during blood collection. In cases where patients had severe dementia, we relied on the Pain Assessment in Advanced Dementia Scale (PAINAD) [22] to identify pain that was not controlled. We included additional variables that might have an association with our results, including the body mass index (BMI), measures of renal function applying the Cockcroft–Gault formula [28], and functionality using the Karnovsky scale [29]. The Mini-Nutritional Assessment scale evaluated the nutritional status [30].

Due to a lack of evidence in published clinical data regarding this issue, it is not possible to firmly assume the size of the changes [31]. We assumed a grouping of two with a 1:1 allocation, an effect size d of 0.8, and a significance alpha of 0.05. With 20 participants in each arm, we had 80% power to detect minimal differences. In addition, the authors expected to lose another 20% of the participants during the follow-up. Therefore, we used a larger sample size to promote a more statistically rigorous method.

### 3.4. Inclusion Criteria

Our study involved patients followed by a specialized palliative care team in the northern region of Portugal with non-oncological illnesses. As previously stated, we only included patients who provided informed consent directly or through their legal representatives.

## 4. Discussion and Conclusions

It is difficult to assess pain in patients receiving palliative or end-of-life care or those with severe dementia, cognitive impairment, or speech disorders, as we cannot solely rely on self-reporting [3,4].

Physical symptoms such as pain, fatigue, and sleep disturbances and psychosocial factors such as stress, coping style, and the availability of emotional support interact in a bidirectional manner and are present at varying levels. These dynamics create a highly personalized disease experience that calls for an individualized, multimodal therapeutic approach [32,33]. Consequently, there is an urgent need to explore non-invasive pain biomarkers that can aid in characterizing pain and customize therapeutic strategies individually.

Several preclinical and clinical studies have investigated the hypothesis that biomarkers can be utilized to identify and quantify pain. A preclinical study shows that inflammatory and neuropathic pain have different biomarkers [34]. Further investigations provided mixed results. For instance, cystatin C levels in cerebrospinal fluid appear to be a predictive marker for postherpetic neuralgia in patients with varicella-zoster virus and a pain marker in women experiencing labor pain. Nevertheless, these biomarkers have no correlation with pain duration or intensity. Studies examining potential biomarkers for chest pain have revealed that cardiac markers utilized for diagnosing cardiac disease and prognosis are associated with tissue damage rather than pain [34].

Previously, many routine peripheral blood parameters have been used as peripheral novel diagnostic, prognostic, and predictive response markers in several diseases [7,8,9,35,36], including dementia [6]. We have recently demonstrated that membrane platelet proteins have some value as pain biomarkers, namely for pain subtype classification and pain intensity characterization [8].

This study assesses whether monocytes, their subtypes, and membrane proteins, such as CD11c, CD86, CD163, and CD206, could serve as non-invasive peripheral biomarkers for identifying and characterizing pain in patients with severe dementia. Our findings suggest that the relative concentrations of monocytes, specifically the percentage of classic monocytes, may serve as valuable biomarkers for pain, irrespective of sex, the presence of dementia, and the type of pain. Moreover, the most evident alterations were found in the levels of CD11c, CD163, and CD206 in classic monocytes, which may contribute to characterizing the pain subtype. We did not detect any changes in CD86 expression related to pain and dementia.

CD11c is a complement receptor often exploited as a single marker to track murine dendritic cells [24]. CD86 is a constitutively expressed phenotypic marker on interdigitating dendritic cells (DCs), Langerhans cells, peripheral blood DCs, memory and germinal center B cells, and macrophages [25]. CD163 is a scavenger receptor for haptoglobin-hemoglobin complexes expressed mainly by monocytes and macrophages induced by inflammatory stimuli [26]. Finally, CD206 is a mannose receptor primarily found on the surface of alternatively activated macrophages and serves as a pattern recognition receptor, contributing to innate and adaptive immunity [27].

This study demonstrated that individuals experiencing pain had higher levels of CD206 and classical monocytes and a lower CD163/CD206 ratio, suggesting the potential use of these parameters as biomarkers for pain. They are independent of gender, dementia, and pain treatment. In fact, in patients with pain, we found a significant increase in the percentage of classic monocytes and the percentage of these cells expressing CD206 and a decrease in the ratio of CD163/CD206 compared to patients without pain. None of these parameters varied with sex, dementia, or treatment with paracetamol and opioids. If these findings are validated in future studies involving a larger number of patients, they should be incorporated into the pain identification process in the peripheral blood of palliative patients with dementia. For patients with well-controlled pain, in comparison with those with non-controlled pain, we did not find statistically significant differences, including in patients with well-controlled pain for seven or more days. These data are relevant because we could expect a reduction in monocyte activation with pain control [13,35,37]. However, it may mean that some underlying mechanisms and causes of pain may be equally active, and that there is only pain desensitization.

Despite the limited sample size, which precluded the assessment of pain types other than nociceptive and mixed pain, we could identify monocyte biomarkers associated with certain types of pain. The levels of CD11c are significantly higher in patients with mixed pain, while the expression levels of CD163 and CD206 are significantly higher in patients with nociceptive pain, with the latter being independent of dementia, as these patients have lower levels of these markers.

Another interesting finding is that the percentage of CD163 in intermediate monocytes showed less expression in patients receiving opioid treatment. Freshly isolated peripheral blood monocytes express a relatively low level of CD163, but this expression increases with the differentiation of such cells into macrophages [26], a role assigned mainly to intermediate monocytes [11,36]. Our results confirm that opioids may substantially affect monocytes’ inactivation, as shown by others [38,39], confirming their anti-inflammatory effect. On the other hand, in patients receiving paracetamol, only an increase in the ratio of CD163/CD206 was detected in non-classic monocytes. According to the literature [40,41,42], there are only references to the increase/activation of CD163 in monocytes frequently related to the liver injury induced by paracetamol. However, we did not find liver disease or dysfunction in these patients.

Monocytes can be used as predictors of dementia, as the relative concentration of total monocytes is higher in patients with dementia, particularly in Alzheimer’s disease and vascular dementia compared with non-demented controls. These findings are in agreement with others [43,44], and the hyperactivation of monocytes has been described as a potential contributor to the progression from mild cognitive impairment to Alzheimer’s disease [45]. While the implication of microglia is well recognized, the exact contribution of peripheral monocytes or macrophages is still largely unknown, especially concerning their role in the various stages of Alzheimer’s disease [45].

Additionally, we did not observe any differences in the levels of monocyte subtypes, contrasting with the findings reported by other authors [45]. However, it should be noted that this report included healthy individuals as controls, in contrast to our study which had only palliative patients, and the controls were those without pain.

Furthermore, the decreased expression levels of CD86 and CD206 could be potential markers of Alzheimer’s disease and vascular dementia, respectively [45]. Conversely, elevated levels of CD163 and CD206 were observed in patients without dementia, suggesting that these markers may serve as negative predictors of dementia. These findings are not in agreement with the literature [45,46,47,48,49,50,51], but they seem to be related with other conditions or are related with the risk of Alzheimer’s disease development, which was not the scope of our study.

Taken together, these findings suggest that the levels of specific monocyte subtypes, namely the classic subtype, along with their associated CD11c, CD163, and CD206 phenotypes may hold value as objective pain biomarkers for palliative patients with dementia. This is a significant advantage as these biomarkers can be quantified without requiring active patient involvement.

The main limitation of our cross-sectional study is the small sample size, which limits the formation of even groups. In this sense, a confirmatory study in a larger cohort of patients should be performed in order to confirm our findings. Additionally, the fact that the pain observed in severe dementia patients can result from different causes and mechanisms, and the fact that patients with pain do not respond similarly to the same treatment may also increase data dispersion. Also, a subtype of dementia may be partially linked to a specific mechanism of neuroinflammation, and therefore this could influence the results [43,44,45,52,53]. Finally, some confounding factors, such as evolution in time and previous treatments, that could affect pain assessment in dementia pain were not explored in this study.

To our knowledge, this is one of the most extensive pain biomarker studies conducted to date and the only study to compare patients with non-oncological pain using specific monocyte biomarkers. Nevertheless, given the small sample size, further studies are warranted to confirm the viability of these markers as indicators of pain and dementia.

For vulnerable and dependent patients who cannot provide a self-report of symptoms, identifying pain biomarkers, such as the ones presented in this study, can aid in adjusting therapeutic strategies in line with the WHO ladder [2], contributing to a faster and more adequate control of pain, and a reduction in associated suffering.

## Figures and Tables

**Figure 1 ijms-24-10723-f001:**
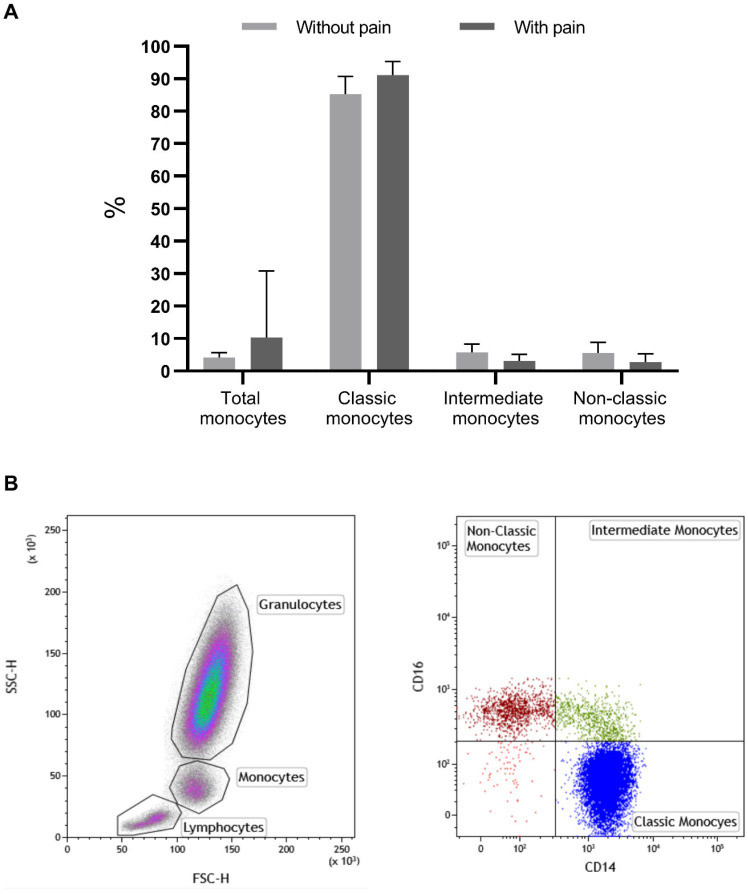
Percentage of monocytes and their subsets, classic, intermediate, and non-classic in patients without and with pain (**A**). In (**B**), there are representative dot-plots of the identification of monocytes based on SSC/FSC (right dot-plot) and of monocyte subsets (left dot-plot): classic monocytes (CD14+/CD16−); intermediate monocytes (CD14+/CD16+); non-classic monocytes (CD14−/CD16+). %—percentage; significant values: total monocytes (10.27% vs. 4.19%, *p* = 0.025); classic monocytes (91.1% vs. 85.3%, *p* = 0.003); intermediate monocytes (5.7% vs. 3.1%, *p* = 0.008); non-classic monocytes (5.53% vs. 2.79%, *p* = 0.011); *p*—significance.

**Figure 2 ijms-24-10723-f002:**
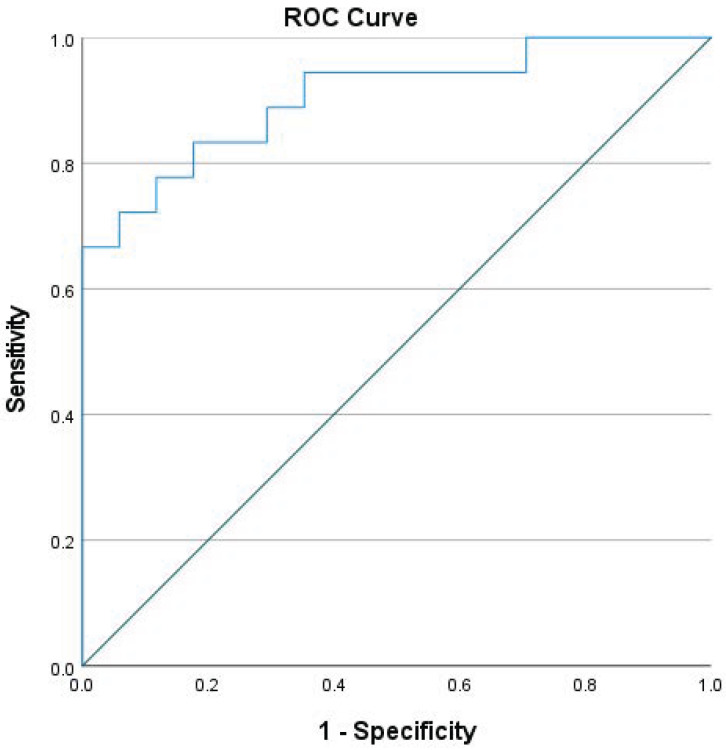
ROC curve of CD11c, CD163, and CD206 as biomarkers of nociceptive pain.

**Table 1 ijms-24-10723-t001:** Characterization of the study population.

Total of Patients: (n = 53)		Patients with Pain and Dementia (n = 38)	Patients with Pain without Dementia (n = 6)	Patients without Pain and without Dementia (Controls) (n = 9)
**Gender**	**Male**	29% (n = 11)	50% (n = 3)	0
	**Female**	71% (n = 27)	50% (n = 3)	100%
**Average age (years)**		84.1	62.5	44.7
**PAINAD scale**	**<5**	15.8% (n = 6)	NA	NA *
	**5–7**	73.7% (n = 28)	NA	NA
	**8–10**	10.5% (n = 4)	NA	NA
**Average numeric pain scale**		NA	4.3	0
**Type of Pain**	**Nociceptive**	39.5% (n = 15)	50% (n = 3)	NA
	**Neuropathic**	2.6% (n = 1)	16.7% (n = 1)	NA
	**Mixed**	57.9% (n = 22)	33.3% (n = 2)	NA
**Type of Dementia**	**Vascular**	36.8% (n = 14)	NA	NA
	**Alzheimer**	36.8% (n = 14)	NA	NA
	**Mixed**	7.9% (n = 3)	NA	NA
	**Other**	18.4% (n = 7)	NA	NA
**Under opioid treatment**		39.5% (n = 15)	66.7% (n = 4)	NA

* NA: Not Applicable. PAINAD: Pain Assessment in Advanced Dementia.

**Table 2 ijms-24-10723-t002:** Monocyte biomarkers and pain.

	Without Pain (n = 9)	With Pain (n = 44)	
	M ± SD	M ± SD	*p*
** *Monocytes* **			
**%**	**4.19 ± 1.45**	**10.27± 20.5**	**0.025 ***
% CD11c	88.52 ± 7.23	294.67 ± 923.03	0.181
MIF 11c	816.98 ± 313.70	820.86 ± 303.38	0.503
% CD86	32.69 ± 12.27	705.21 ± 3073.33	0.130
MIF 86	634.20 ± 20.58	608.65 ± 143.02	0.850
% CD163	85.63 ± 6.36	115.35 ± 150.62	0.487
**MIF 163**	**3061.99 ± 1033.45**	**2378.85 ± 1840.89**	**0.050 ***
**% CD206**	**14.08 ± 7.80**	**59.36 ± 230.35**	**0.047 ***
**MIF 206**	**534.29 ± 54.31**	**469.01 ± 14.66**	**0.019 ***
% 11c/86	31.76± 12.10	99.96 ± 352.47	0.123
**% 163/206**	**9.5 ± 5.7**	**4.8 ± 5.6**	**0.004 ****
** *Classical monocytes* **			
**%**	**85.3 ± 5.4**	**91.1 ± 4.2**	**0.003 ****
% CD11c	91.9 ± 8.3	93.6 ± 9.4	0.250
MIF 11c	740.6 ± 286.2	825.4 ± 260.7	0.182
% CD86	30.1 ± 12.3	21.7 ± 13.7	0.070
MIF 86	592.1 ± 13.7	593.1 ± 23.8	0.781
**% CD163**	**95.6 ± 6.3**	**88.0** ± **9.4**	**0.010 ****
MIF 163	3185.1 ± 1119.8	2526.7 ± 1799.5	0.079
**% CD206**	**11.2 ± 6.9**	**6.7 ± 6.3**	**0.039 ***
**MIF 206**	**503.4 ± 48.6**	**472.9 ± 89.8**	**0.021 ***
% 11c/86	29.0 ± 12.1	21.0 ± 13.0	0.091
**% 163/206**	**11.1 ± 6.8**	**6.5 ± 6.1**	**0.038 ***
** *Intermediate* **			
**%**	**5.7 ± 2.6**	**3.1 ± 2.0**	**0.008 ****
% CD11c	98.9 ± 1.0	98.4 ± 3.1	0.770
MIF 11c	1769.1 ± 601.1	1743.8 ± 551.8	0.947
% CD86	78.0 ± 9.5	68.5 ± 22.5	0.376
MIF 86	813.7 ± 90.3	853.4 ± 103.1	0.174
% CD163	88.1 ± 9.6	79.7 ± 13.9	0.064
MIF 163	2397.9 ± 1125.5	2843.9 ± 3818.2	0.174
% CD206	43.7 ± 17.6	35.7 ± 15.5	0.328
MIF 206	537.4 ± 70.1	548.4 ± 185.5	0.682
% 11c/86	77.8 ± 9.6	68.4 ± 22.4	0.362
% 163/206	41.5 ± 18.0	33.2 ± 15.3	0.229
** *Non-classical* **			
**%**	**5.53** ± **3.25**	**2.79 ± 2.50**	**0.011 ***
% CD11c	78.1 ± 30.4	78.7 ± 26.2	0.812
MIF 11c	1612.8 ± 623.4	1713.7 ± 548.2	0.518
% CD86	58.8 ± 26.0	64.2 ± 26.9	0.391
MIF 86	791.0 ± 82.1	854.3 ± 121.9	0.129
% CD163	14.6 ± 12.4	12.6 ± 7.6	0.771
MIF 163	813.05 ± 196.72	885.05 ± 327.74	0.947
% CD206	17.0 ± 16.3	11.4 ± 9.2	0.187
MIF 206	634.7 ± 203.5	572.9 ± 347.5	0.099
% 11c/86	58.7 ± 26.0	64.1 ± 26.9	0.383
% 163/206	4.1 ± 4.7	3.8 ± 4.9	0.561

M—mean; SD—standard deviation * *p* ≤ 0.05 ** *p* ≤ 0.01. %—percentage; MFI—mean fluorescence intensity; *p*: significance. The statistically significant differences are highlighted in bold.

**Table 3 ijms-24-10723-t003:** Type of pain and monocyte biomarkers.

	Nociceptive (n = 18)	Mixed (n = 20)	
	M ± SD	M ± SD	*p*
Granulocytes	71.75 ± 12.42	73.00 ± 11.06	0.808
Linfocytes	20.51 ± 10.86	637.36 ± 1843.35	0.395
** *Monocytes* **			
%	5.48 ± 2.26	15.77 ± 29.48	0.301
**% CD11c**	**83.16 ± 20.92**	**539.46 ± 1332.28**	**0.002 ****
MIF 11c	743.56 ± 165.92	870.15 ± 362.37	0.068
% CD86	24.11 ± 13.78	1494.07 ± 4445.82	0.649
MIF 86	639.67 ± 37.13	574.51 ± 204.14	0.466
% CD163	82.39 ± 9.20	153.51 ± 217.85	0.671
**MIF 163**	**3010.81 ± 2422.58**	**1698.99 ± 793.94**	**0.013 ****
% CD206	8.42 ± 5.07	118.12 ± 333.30	0.605
**MIF 206**	**510.91 ± 54.65**	**425.07 ± 197.91**	**0.000 *****
% 11c/86	23.16 ± 12.93	189.30 ± 510.08	0.605
**% 163/206**	**5.42 ± 3.46**	**4.24 ± 7.72**	**0.027 ***
** *Classical monocytes* **			
%	89.68 ± 4.39	92.26 ± 2.55	0.092
**% CD11c**	**91.57 ± 9.69**	**97.88 ± 2.13**	**0.005 ****
**MIF 11c**	**709.87 ± 157.26**	**928.22 ± 249.20**	**0.008 ****
% CD86	21.13 ± 13.76	22.16 ± 14.10	0.779
MIF 86	586.91 ± 15.35	598.80 ± 27.90	0.291
% CD163	89.06 ± 9.86	86.85 ± 9.09	0.269
**MIF 163**	**3053.45 ± 2405.70**	**1905.96 ± 555.69**	**0.041 ***
% CD206	6.79 ± 4.50	6.18 ± 7.92	0.166
**MIF 206**	**478.14 ± 39.38**	**469.28 ± 129.75**	**0.016 ****
% 11c/86	20.14 ± 12.67	22.05 ± 14.04	0.741
% 163/206	6.64 ± 4.19	5.98 ± 7.64	0.137
** *Intermediate* **			
%	3.77 ± 2.37	2.77 ± 1.27	0.290
**% CD11c**	**97.99 ± 2.65**	**99.66 ± 0.39**	**0.004 ****
**MIF 11c**	**1518.99 ± 474.10**	**1976.88 ± 536.37**	**0.013 ***
% CD86	66.96 ± 26.87	70.89 ± 17.14	0.974
MIF 86	832.39 ± 98.82	871.68 ± 104.54	0.488
% CD163	80.41 ± 15.61	79.65 ± 13.11	0.644
MIF 163	3678.20 ± 5385.99	1849.45 ± 629.27	0.488
% CD206	36.37 ± 16.27	33.21 ± 15.31	0.668
**MIF 206**	**575.94 ± 124.37**	**517.74 ± 247.86**	**0.041 ***
% 11c/86	66.79 ± 26.82	70.88 ± 17.13	0.974
% 163/206	34.37 ± 16.14	30.24 ± 15.10	0.530
** *Non-classical* **			
**%**	**3.72 ± 2.69**	**1.82 ± 1.09**	**0.037 ***
% CD11c	74.93 ± 30.12	87.43 ± 18.27	0.409
**MIF 11c**	**1522.18 ± 466.66**	**1999.55 ± 447.14**	**0.005 ****
% CD86	60.33 ± 31.01	73.32 ± 17.57	0.276
MIF 86	828.17 ± 104.68	864.42 ± 98.72	0.269
% CD163	13.68 ± 7.98	12.84 ± 7.77	0.520
MIF 163	941.04 ± 357.84	775.93 ± 89.13	0.276
% CD206	12.46 ± 10.10	10.40 ± 7.58	0.869
MIF 206	729.17 ± 465.05	443.38 ± 70.24	0.060
% 11c/86	60.24 ± 31.02	73.22 ± 17.65	0.283
% 163/20	4.45 ± 4.52	3.54 ± 5.76	0.209

M—mean; SD—standard deviation; * *p* ≤ 0.05 ** *p* ≤ 0.01 *** *p* ≤ 0.001. %—percentage; MFI—mean fluorescence intensity; *p*: significance. The statistically significant differences are highlighted in bold.

**Table 4 ijms-24-10723-t004:** Differences between CD163 values in patients receiving opioid therapy and in patients without opioids.

		Without Opioids	Opioid Therapy	
		M ± SD	M ± SD	*p*
**Monocytes**	%163	126.57 ± 169.27	80.54 ± 9.92	0.135
	MIF163	2645.37 ± 2041.58	2246.61 ± 985.25	0.441
** *Classical monocytes* **				
	%163	91.1 ± 8.1	86.6 ± 10.7	0.108
	MIF163	2858.3 ± 1978.5	2303.3 ± 1064.9	0.279
** *Intermediate monocytes* **				
	%163	85.7 ± 9.2	74.0 ± 16.3	0.010 *
	MIF163	3363.0 ± 4288.1	1755.8 ± 440.3	0.051
** *Non-classical monocytes* **				
	%163	13.9 ± 10.1	11.4 ± 5.1	0.334
	MIF163	936.97 ± 362.22	762.52 ± 128.76	0.056

M—mean; SD—standard deviation; * *p* ≤ 0.05. %—percentage; MFI—mean fluorescence intensity; *p*: significance.

**Table 5 ijms-24-10723-t005:** Monocyte biomarkers and dementia.

	Without Dementia	With Dementia		
	M ± SD	M ± SD		*p*
**Monocytes**				
**%**	**4.73 ± 2.11**	**11.08**	**22.11**	**0.037 ***
% CD11c	88.29 ± 10.05	330.11 ± 996.73		0.125
MIF 11c	784.79 ± 256.61	835.32 ± 321.77		0.305
% CD86	29.65 ± 14.25	821.8 ± 3319.12		0.271
MIF 86	631.93± 24.31	605.24 ± 154.41		0.791
% CD163	85.39 ± 7.11	120.54 ± 162.75		0.335
**MIF 163**	**3420.37 ± 2557.46**	**2108.14 ± 1055.71**		**0.032 ***
% CD206	12.68 ± 9.22	67.72 ± 248.83		0.138
**MIF 206**	**562.85± 129.08**	**445.58 ± 123**		**0.001 *****
% 11c/86	28.83 ± 13.73	112.91 ± 380.71		0.236
**% 163/206**	**9.2 ± 8.5**	**4.1 ± 3.2**		**0.010 ****
** *Classical monocytes* **				
%	87.8 ± 5.6	91 ± 4.4		0.054
% CD11c	91.1 ± 10.3	94.3 ± 8.6		0.097
MIF 11c	726 ± 234.2	847.4 ± 272.2		0.081
% CD86	27.5 ± 13.9	21.4 ± 13.4		0.114
MIF 86	591.7 ± 11.9	593.5 ± 25.7		0.991
**% CD163**	**93.6 ± 8.2**	**87.5 ± 9.3**		**0.012 ***
MIF 163	3497.5 ± 2534.9	2265 ± 978.4		0.063
% CD206	10.6 ± 9.2	6.1 ± 4.5		0.087
**MIF 206**	**528.2 ± 127**	**456.1 ± 40.6**		**0.002 ****
% 11c/86	26.5 ± 13.3	20.6 ± 12.8		0.114
% 163/206	10.4 ± 9	6 ± 4.3		0.089
** *Intermediate* **				
**%**	**4.7 ± 2.5**	**3.1 ± 2.1**		**0.028 ***
% CD11c	98.6 ± 2.1	98.4 ± 3.2		0.832
MIF 11c	1653.2 ± 565.9	1791.9 ± 552.9		0.417
% CD86	71.5 ± 20.5	69.7 ± 21.4		0.920
MIF 86	820.3 ± 98.1	857.6 ± 101.8		0.158
**% CD163**	**87.4 ± 10**	**78.5 ± 14.1**		**0.021 ***
MIF 163	3726.8 ± 5665.9	2320.9 ± 1730.1		0.085
% CD206	42.5 ± 15.9	34.8 ± 15.8		0.213
MIF 206	607.6 ± 226.6	518.5 ± 130.7		0.107
% 11c/86	71.4 ± 20.4	69.6 ± 21.4		0.903
% 163/206	40.5 ± 15.9	32.1 ± 15.6		0.095
** *Non-classical* **				
%	4.27 ± 3.41	2.87 ± 2.46		0.221
% CD11c	76.5 ± 30	79.5 ± 25.5		0.656
MIF 11c	1636.6 ± 577.9	1721.2 ± 554.9		0.601
% CD86	61.5 ± 27.9	64 ± 26.3		0.648
MIF 86	828.5 ± 94.4	848.7 ± 127.4		0.664
% CD163	13.3 ± 9.7	12.9 ± 8.2		0.764
MIF 163	870.87 ± 339.24	871.86 ± 296.75		0.714
% CD206	14.6 ± 13.8	11.5 ± 9.3		0.368
**MIF 206**	**752.9 ± 447.3**	**507.9 ± 218.5**		**0.008 ****
% 11c/86	61.4 ± 27.9	63.9 ± 26.3		0.640
% 163/206	3.8 ± 4	3.9 ± 5.2		0.490

M—mean; SD—standard deviation; * *p* ≤ 0.05 ** *p* ≤ 0.01 *** *p* ≤ 0.001. MFI—mean fluoresce intensity; *p*: significance. The statistically significant differences are highlighted in bold.

## Data Availability

The data presented in this study are available on request from the corresponding author.

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
