# Peer review of "Monocytes in the Characterization of Pain in Palliative Patients with Severe Dementia—A Pilot Study"

_ijms, 2023, doi:10.3390/ijms241310723_

Round 1
Reviewer 1 Report
In this study the authors investigated whether circulating monocytes and related membrane proteins identified as a cluster of differentiation, could be potential non-invasive peripheral biomarkers in identifying and characterizing pain in patients with severe dementia. The topic is interesting. Some concerns and suggestions are listed as below:
Seventeen patients were excluded from blood sample collection due to their fragile condition, hypovolemia, and difficult venous access. The number is high. Some may arugue that this may cause selection bias.
It is accepted that women tend to report more pain than men. Sex differences (pain or/and monocyte subsets) should not be ignored in this study.
The number of control group is low (including patients with pain without dementia).
Males should be included in the control group.
In table 1, average ages were not matched among groups.
How these patients were treated? Some may argue that some treatments may have effects on final results.
FACS scatter plots (related membrane proteins ) of each group should be provided (Figue 1).
Apart from the percentages in Figure1, the numbers of each monocyte subset should be provided.
In Figure 1, no bars in non-classical monocytes?
It is not clear for readers if changes of circulating monocytes and related membrane proteins have any correlations with clinical features.
The sensitivity and specificity of circulating monocytes and related membrane proteins as potential biomarkers were lacking in this study.
Functions of these monocytes (or monocyte subset) should be investiagted. Additional experiments are needed.
How about CX3CR1 and CCR2 expression?
Author Response
Replies to the reviewers
We thank both reviewers for their comments and suggestions. Their suggestions improved the quality of the manuscript.
Please find our replies below.
Reviewer 1
- Seventeen patients were excluded from blood sample collection due to their fragile condition, hypovolemia, and difficult venous access. The number is high. Some may argue that this may cause selection bias.
Answer: Thanks for your comment. Patients were blindly and randomly selected from a pool of patients followed by a palliative care team, and it is recognized that collecting blood samples at the end of life is difficult. As far as we know, this is one of the largest studies in this area with palliative patients. Please note that we did not include healthy participants. The control group consists of patients that present no pain or dementia but have advanced, progressive, and fatal non-oncological diseases. Although this control group has fewer patients, the findings are significant using adequate statistical tests.
- It is accepted that women tend to report more pain than men. Sex differences (pain or/and monocyte subsets) should not be ignored in this study.
Answer: Our study found no differences in the prevalence of pain in males and/or females. As the reviewer recommended, we included and highlighted in yellow the sentence, “We had no significant differences between the prevalence of pain in males and/or females” (page 4, lines 188-189). However, we found significant differences in the percentage of cells expressing CD86 and in the ratio CD11c/CD86 in patients with pain according to sex, being higher in men, as mentioned previously on page 8, lines 236-240 of the manuscript. So, these markers seem to be sex-dependent.
- The number of control group is low (including patients with pain without dementia). Males should be included in the control group. In table 1, average ages were not matched among groups.
Answer: This is a laboratory study in palliative patients, particularly those under clinical follow-up by a specialized palliative care team. We stress again that this is the largest study with patients with these characteristics, and we did not include healthy participants. The control group patients present no pain or dementia but have advanced, progressive, and fatal diseases (non-oncological).
Although this control group has a smaller number of patients, the findings are significant using adequate statistical tests. Further, as we stated before, “We had no significant differences between the prevalence of pain in males and/or females” (page 4, lines 188-189)”.
- How these patients were treated? Some may argue that some treatments may have effects on final results.
Answer: As the reviewer recommended, we included and highlighted in yellow (page 12, lines 342-349) that “patients under opioids, when compared with patients that did not receive opioids, did not present statistically significant differences in the percentage of monocytes and in the monocytes subsets (monocytes expressing different characteristic CDs). We had identical findings when comparing patients with other analgesics, such as paracetamol and/or NSAIDs, with patients without any pain treatment.”
- FACS scatter plots (related membrane proteins ) of each group should be provided (Figue 1). Apart from the percentages in Figure1, the numbers of each monocyte subset should be provided. In Figure 1, no bars in non-classical monocytes?
Answer: We thank the reviewer for this suggestion. We have included in Figure 1 representative dot plots of the evaluation of monocytes and of the different monocytes subsets. We also included bars in non-classical monocytes.
- It is not clear for readers if changes of circulating monocytes and related membrane proteins have any correlations with clinical features.
Answer: As the reviewer has recommended, we included and highlighted in yellow (page 4, lines 189-191) that clinical features such as nutritional status, height, weight, renal function, and hepatic function, which have an important contribution for pharmacokinetics, do not have statistically significant differences for monocyte expression and/or existence and types of pain.
- The sensitivity and specificity of circulating monocytes and related membrane proteins as potential biomarkers were lacking in this study.
Answer: Thanks for this comment. With a multilogistic regression we did not find any statistically relevant differences, so we did not do a multinominal regression. The specificity of the model is 88.2%, and the sensibility is 77.8%. The model’s discriminative capacity is excepcional (0.905), with the area under the curve being statistically significant. As you recommended, we added and highlighted in yellow this information (page 9-10, lines 263-266) and we added Figure 2 (“ROC curve of CD11c, CD163 and CD206 as biomarkers of nociceptive pain”).
- Functions of these monocytes (or monocyte subset) should be investigated. Additional experiments are needed. How about CX3CR1 and CCR2 expression?
Answer: We agree with the need of more investigation in this field. However, this study aims to assess the levels of monocytes subsets (classical, non-classical, and intermediate) identified by the transmembrane proteins CD11c, CD86, CD163, and CD206 in order to identify the potential of these membrane proteins as pain biomarkers characterization in patients who cannot self-report their pain. CX3CR1 and CCR2 have been evaluated recently by other authors in chronic pain and neuropathic pain (Montaque-Cardoso et al., 2020; Rita silva et al., 2021 ). We thank the reviewer for the input and will try to evaluate CX3CR1 and CCR2 expression in future investigations.

Reviewer 2 Report
The authors investigated monocytes and their clusters of differentiation as potential peripheral biomarkers in identifying and characterizing pain in patients with severe dementia. The article is intruiguing and original, however it should to be improved prior to pubblication in the Journal. In the methods' section the authors should clarify the criteria for the diagnosis of dementia and also the work-out to identificate the different forms of dementia and the severity (degenerative, vascular..., risk factors, family hystory, years of disease, other neurological signs or symtoms, neuroimaging, neurocognitive evaluation...). Maybe these data could add fundamental information to this translate research to clinical settings.
Author Response
Replies to the reviewers
We thank both reviewers for their comments and suggestions. Their suggestions improved the quality of the manuscript.
Please find our replies below.
Reviewer 2
- In the methods' section the authors should clarify the criteria for the diagnosis of dementia and also the workout to identificate the different forms of dementia and the severity (degenerative, vascular..., risk factors, family history, years of disease, other neurological signs or symptoms, neuroimaging, neurocognitive evaluation...). Maybe these data could add fundamental information to this translate research to clinical settings.
Answer: Thanks for your comment. As we stated on page 3, line 99 of the manuscript, we obtained information about the type of dementia, and our patients were followed by a specialized palliative care team at home (page 4, line 168). Most of these patients presented a diagnosis of severe dementia, although we had no information concerning the form of dementia. We stress that the study aimed to assess the levels of monocytes subsets (classical, non-classical, and intermediate) identified by the transmembrane proteins CD11c, CD86, CD163 and CD206 in order to identify the potential of these membrane proteins as pain biomarkers characterization in patients who cannot self-report their pain.

Round 2
Reviewer 1 Report
The authors did not address previous concerns by providing additional data. For example, the authors did not address how the sample size was calculated since the control number is low.
Author Response
Thanks for your comment. Although this control group has fewer patients, the findings are significant using adequate statistical tests. Due to lack of evidence in published clinical data regarding this issue, it is not possible to firmly assume the size of changes . Assuming a grouping of two with a 1:1 allocation, an effect size d of 0.8, and a significance alpha of 0.05. With 20 participants in each arm, we had 80% power to detect minimal differences. In addition, the authors expected to lose another 20% of the participants during the follow-up. Therefore, we used a larger sample size to promote a more statistically rigorous method. We introduced and highlighted this explanation in page 4, lines 167-172.

Reviewer 2 Report
I suggest to include the concept raised in my previous comments in the limitations of the study. In particular, as highlighted by recent data about the role of neuroinflammation in the different form of dementia (PMID: 33318676, PMID: 35682903...), a subtype of dementia may be partially linked to a specific mechanisms of neuroinflammation and it could pour over the results.
Author Response
Thanks for your comment. We agree with your concern and included and highlighted it in page 15, lines 467-468. Previously, in page 14, lines 436-448, we had some concerns about this issue.

Round 3
Reviewer 1 Report
The authors have answered my major concerns.